# Enabling high-throughput biology with flexible open-source automation

Emma J Chory[1,2,3,†] (ID), Dana W Gretton[1,†,‡] (ID), Erika A DeBenedictis[1,4] (ID) & Kevin M Esvelt[1,*] (ID)

## Abstract

Our understanding of complex living systems is limited by our capacity to perform experiments in high throughput. While robotic systems have automated many traditional hand-pipetting protocols, software limitations have precluded more advanced maneuvers required to manipulate, maintain, and monitor hundreds of experiments in parallel. Here, we present Pyhamilton, an open-source Python platform that can execute complex pipetting patterns required for custom high-throughput experiments such as the simulation of metapopulation dynamics. With an integrated plate reader, we maintain nearly 500 remotely monitored bacterial cultures in log-phase growth for days without user intervention by taking regular density measurements to adjust the robotic method in real-time. Using these capabilities, we systematically optimize bioreactor protein production by monitoring the fluorescent protein expression and growth rates of a hundred different continuous culture conditions in triplicate to comprehensively sample the carbon, nitrogen, and phosphorus fitness landscape. Our results demonstrate that flexible software can empower existing hardware to enable new types and scales of experiments, empowering areas from biomanufacturing to fundamental biology.

**Keywords** bioautomation; high-throughput biology; liquid-handling; robotics; systems biology

**Subject Categories** Biotechnology & Synthetic Biology; Methods & Resources

**Mol Syst Biol. (2021) 17: e9942**

## Introduction

Comprehensive, well-replicated experiments are foundational to rigorous science, but humans can only perform so many actions simultaneously. One possible solution is automation, which has been widely implemented in biotechnology (Sparkes *et al*, 2010; Appleton *et al*, 2017; Freemont, 2019) to facilitate routine tasks involved in DNA sequencing (Meldrum, 2000), chemical synthesis (Ley *et al*, 2015), drug discovery (Schneider, 2018), and molecular biology (Smanski *et al*, 2014). In principle, flexibly programmable robots could enable diverse experiments requiring conditions and replicate numbers beyond the capabilities of human researchers across a range of disciplines (Vasilev *et al*, 2011; Hans *et al*, 2018; Keller *et al*, 2019). However, existing software for liquid-handling robots focuses narrowly on automating protocols designed for hand pipettes, while foundry languages such as Antha and remote labs such as Emerald Cloud focus on automating workflows rather than expanding experimental limits. As such, even labs with well-established high-throughput infrastructures struggle to utilize the full potential of their robots, precluding many complex experiments that require flexible programming (Appleton *et al*, 2017).

Bioautomation lags behind the advancing field of manufacturing, where robots are expected to be task-flexible, responsive to new situations, and interactive with humans or remote management systems when ambiguous situations or errors arise (Appleton *et al*, 2017). A key limitation is the lack of a comprehensive, suitably abstract, and accessible software ecosystem (Bär *et al*, 2012; Linshiz *et al*, 2014; Walsh *et al*, 2019). Though bioinformatics is increasingly open-sourced (Gentleman *et al*, 2004; Cock *et al*, 2009), bioautomation has been slow to adopt key practices such as modularity, version control, and asynchronous programming. To enable flexible high-throughput experimentation, we developed Pyhamilton, a Python package that facilitates high-throughput operations within the laboratory, with protocols that can be easily shared and modified. Further, Pyhamilton allows liquid-handling robots to execute previously unimaginable and increasingly impressive methods. With this package, users can run robot simulations to troubleshoot and plan experiments, schedule experimental processes, implement error handling for quick troubleshooting, and easily integrate robots with external equipment.

## Results

Pyhamilton enables Hamilton STAR, STARlet, and VANTAGE liquid-handling robots to be programmed using Python. This allows

1   Media Laboratory, Massachusetts Institute of Technology, Cambridge, MA, USA
2   Institute for Medical Engineering and Science, Massachusetts Institute of Technology, Cambridge, MA, USA
3   Broad Institute of MIT and Harvard, Cambridge, MA, USA
4   Department of Biological Engineering, Massachusetts Institute of Technology, Cambridge, MA, USA
    *Corresponding author. Tel: +1 617 715 2615; E-mail: esvelt@mit.eduE-mail: esvelt@mit.edu
    †These authors contributed equally to this work
    ‡All correspondence regarding Pyhamilton software development should be directed to DWG: (dgretton@mit.edu, https://github.com/dgretton/).

for robotic method development to benefit from standard software paradigms, including exception handling, version control, object-oriented programming, and other cornerstone computer science principles (Table EV1, Movie EV1). Pyhamilton seamlessly connects with Hamilton robots (Appendix Fig S1), can interface with custom peripherals (Fig 1A), and contains unique Python classes corresponding to robotic actions (i.e. aspirate and dispense) and consumables (i.e. plates and pipette tips) (See Dataset EV1). To enable method troubleshooting, Pyhamilton can also simulate methods through Hamilton run control software (Movies EV2–EV4) and incorporate any Python package (i.e. enabling error notifications via push, text message, or Slack). Finally, in addition to the functionalities we present, researchers can now also develop their own flexible code that may be useful for increasingly specialized applications.

### Enabling improved throughput of basic robotic tasks

Complex procedures are built from simple tasks, but the capabilities of a pipetting robot are limited by standard liquid-handling software. For example, an 8-channel head cannot be readily programmed to pipette into two 24-well plates simultaneously, although doing so is physically possible (Fig 1B). This limits many high-throughput assays: automation of methods involving 24-well plates is no faster than hand-pipetting, since robots and researchers pipette one plate at a time. Thus, we first demonstrate that Pyhamilton easily enables pipetting of liquids over two 24-well plates simultaneously (Fig 1B and C, Table EV2), thereby doubling the speed (Movie EV5). This can be critical for bacterial assays involving heated liquid agar which solidifies quickly. This simple example demonstrates the advantages of making full use of the robot's mechanical capabilities, freed from software constraints.

### Enabling liquid transfers requiring complex calculations

Despite having far greater physical capabilities than a fixed-volume multichannel pipette, it is difficult to implement complex liquid transfer patterns involving different volumes on a robot because programming using standard software is prohibitively monotonous (Movie EV1). The ability to faithfully execute experiments involving hundreds of different pipetting volumes could enable new types of applications such as evolutionary dynamics experiments examining gene flow (Slatkin, 1987), population symbiosis (Kaneko & Ikegami, 1992), sources and sinks (Dias, 1996), genetic drift (Lande, 1976; Gillespie, 2000), and the spread of gene drive systems (Esvelt *et al*, 2014; Noble *et al*, 2017) (Fig 1D). We accordingly used Pyhamilton to enable the flexible transfer of organisms between populations in a 96-well plate, using pre-programmed migration rates to simulate geographic barriers (Fig 1E).

A human would have great difficulty performing or programming hundreds of variable pipetting actions in many directions, in any reasonable time frame, without errors. With Pyhamilton, simple abstractions and data structures make this task straightforward. Instead of exhaustively specifying each pipetting step, we specified liquid transfer patterns as matrices and allowed Python to compile the requisite steps. We demonstrate liquid transfer to nearby plates and between adjacent wells to model "flow" or "diffusion" across the miniaturized landscape of a 96-well plate. We then simulate genetic flow by visualizing the point spread of a drop of dye near the center of a plate (Fig 1F and G, Table EV2). The amount of liquid exchanged and the number of wells is arbitrary, defined as a sparse matrix where the rows are source wells, the columns are destination wells, and the values are the fraction of liquid transferred (Appendix Fig S2). Each iteration, the robot performs several hundred bi-directional liquid transfers to apply the matrix operations (Movie EV6). Succinct code (Fig 1G) can generate both symmetric and asymmetric diffusion patterns, which could be combined with a phenotypic reporter to experimentally simulate arbitrarily directionally bounded or unbounded migration (Fig 1D) with many model organisms such as *E. coli*, yeast, or even nematodes.

### Enabling feedback control to maintain culture conditions

Though most liquid-handling robots are used to execute a list of precompiled instructions (e.g., assembling reagents for many PCRs), many potential applications require making real-time modifications in response to changing data. For example, a turbidostat is a culture of cells that is maintained at a constant density by making real-time adjustments to the flow rate of media based on turbidity sensing. In practice, this is accomplished with process controls which measure the optical density (OD) of a culture *in situ* (Horinouchi *et al*, 2014; Haby *et al*, 2019). However, turbidity probes are both costly and not amenable to very high-throughput experiments (Takahashi *et al*, 2017; Wong *et al*, 2018; Hemmerich *et al*, 2018). Thus, we sought to

---

**Figure 1. Example Pyhamilton applications.**

A  Generalizable Python outline for writing custom Pyhamilton code to interface with robot and integrated equipment such as plate readers (e.g., ClarioStar) and custom pump arrays.

B  Expanded robot capabilities allow for improved throughput of laboratory assays across 24-well plates.

C  Example code required to run a bacterial assay across multiple simultaneous plates. Code for bacteriophage plaque assay is shown (see supplemental methods).

D  Implementing complex and arbitrary bi-directional liquid handling to simulate experiments such as unbounded (left) or bounded (right) population flow across a geographic region, such as a river.

E  Geographic "barriers" described in matrix format

F  Simulation of bounded and unbounded migration (top), and visualization of the liquid patterns executed by the robot each iteration (bottom). Solid blue box designates "high" geographic barrier, dashed blue box designates a "medium" geographic barrier.

G  Example code required to run population dynamics simulations, using a sparse matrix to assign source wells, destination wells, and volume transfer fractions.

H  Real-time monitoring of on-deck turbidostats enables feedback control to equilibrate cultures to a set density.

I  Plate reader measurements for OD (top), and respective estimated growth estimates (bottom) obtained from data from 24 replicates. Data are smoothed with rolling mean and outlier points are excluded. OD set-point shown in red.

J  Example code required to maintain on-deck turbidostats using a transfer function to calculate k-estimates and volume transfer rates.

**A   GENERAL METHOD OUTLINE**

```
def main():
    ... ## insert each method code here (i.e. Fig 1C, 1F, 1H, etc)
with HamiltonInterface() as ham_int, ClarioStar() as reader_int, LBPumps() as pump_int:   ## load Hamilton, plate reader, and pumps
    sys_state.instruments = ham_int, reader_int, pump_int    ## Define instruments used in experiment
    system_initialize()    ## initialize robot
    main()    ## run method
```

**B   EXPANDED ROBOTIC CAPABILITIES:  Improved Throughput of Assays**

Culture-Based Assay   Manufacturer Software   Pyhamilton

**C   BACTERIAL ASSAY CODE**

```
## Service two 24-well plates at once
for assay_plates in get_2_plates():
    ## Define assay reagents and destination wells
    reagents, bacteria, destination_wells = \
        get_8_wells(assay_plates)
    ## Perform neccessary pipetting steps
    prepare_assays(reagents, bacteria, \
        destination_wells)
```

**D   COMPLEX LIQUID HANDLING:  Population Dynamics**

Population flow   Bounded population flow   Number per county (Normalized to Area)

Mississippi River

t = 0   t = 100   t = 0   t = 100

**E**

Symmetric   Asymmetric   Geographic Barrier

None / Medium / High

**F**

Symmetric Flow Pattern
Simulation / Execution

Asymmetric Flow Pattern
Simulation / Execution

**G   POPULATION  DYNAMICS CODE**

```
## Move dye from one plate or well to the next (in loop)
for source_plate, destination_plate in zip(plates, plates[1:]):
    ## Service one column at a time
    for column in groups_of_8(range(96)):
        ## Calculate liquid transfer volumes
        transfers = nonzero_transfers(flow_matrix,
            source_plate, column)
        ## Perform pipetting steps (in loop)
        for destinations, volumes in transfers:
            load_new_tips() # pick up tips
            ## Aspirate liquid from source wells
            aspirate_from(select_channels(source_plate, \
                column, destinations), volumes)
            ## Dispense liquid to all destinations
            dispense_to(destinations, volumes)
            ## Return tips to housing unit
            put_tips_back()
```

**H   FEEDBACK-RESPONSIVE ADJUSTMENTS:  On-Deck Turbidostats**

Real-time Monitoring
Density / Fluor. / Lum. — Time

Feedback Adjustment
Flow Rate / K-estimate / Transfer Volume — Time

Turbidostat — Measure OD → Dilute → Remove waste

**I**

Absorbance [AU] vs Time (hours)
Growth rate Estimate (K) vs Time (hours)

**J   TURBIDOSTAT METHOD CODE**

```
timer = Timer()                          # Start time
while True:
    timer.start(cycle_time)              # Iteration frequency
    sample_turbs()                       # Move plate to reader
    readings = read_ods()                # Take plate readings
    # Calculate K-estimates and replacement volumes
    replace_volumes = transfer_function(readings)
    # Replace media based on K-estimates
    replace_media(replace_volumes)
    system_clean()                       # Clean system
    timer.wait()
```

**Figure 1.**

leverage the flexibility of Pyhamilton to multiplex the maintenance of many bacterial turbidostats by adjusting the volume of liquid transfers in response to real-time density measurements obtained using an integrated plate reader (Fig 1H–J). The method equilibrates each culture, growing in a multiwell microplate, to a setpoint (Fig 1 I) in response to these measurements by applying a transfer function to calculate the growth rate (k-value) and adjustment volume for each individual well over time (Fig 1J).

**Asynchrony enables high-throughput turbidostats**

To maximize the number of turbidostats that can be maintained, we next developed a more complex method which uses asynchronous programming to execute multiple robotic steps simultaneously— in this case plate reading and pipetting (Appendix Fig S4). This allows for up to 480 cultures to be maintained with real-time fluorescent reporter monitoring on a single small robot, nearly 20× more than

can be readily achieved with multiplexed mini-bioreactor setups (Hans *et al,* 2018; Haby *et al,* 2019). In this method, bacterial cultures are inoculated into 96-well clear-bottom plates and their ODs and fluorescence levels are measured with an integrated plate reader (Fig 2A, Movie EV7). To minimize waste, consumables, and prevent media contamination, we also implemented a cleaning process (Fig 2A): after each media transfer, each tip is sterilized with 1% bleach, rinsed in water, and returned to its housing unit (Fig 2A, Table EV2). To further minimize the possibility of cross-contamination between wells, each culture is assigned its own tip and media reservoir by housing replenishing media within high-volume 96-well plates. We confirmed that this method introduces no measurable cross-contamination by inoculating 96 turbidostats with four different bacterial cultures

expressing RFP, YFP, CFP, or no fluorescent protein in a grid-like pattern with no-bacteria controls (Fig 2C). We then monitored the absorbance and fluorescence levels in real time and maintained the cultures at OD 0.8 for 24 h. We observed no cross-contamination and no growth in the no-bacteria controls during this time (Fig 2 C). We also inoculated the same bacterial strains at 6 different starting densities (OD = 0.0–0.8) and demonstrated that irrespective of initial conditions, the feedback control algorithm equilibrates each culture to its set point within 12 h (Fig 2D). Finally, we confirmed that the method could support culture maintenance of bacteria with varying growth rates (Appendix Fig S5A–C), with no measurable back-contamination of media (Appendix Fig S5D), for up to 2 days without experimenter intervention (Appendix Figs S5E and F, and S6).

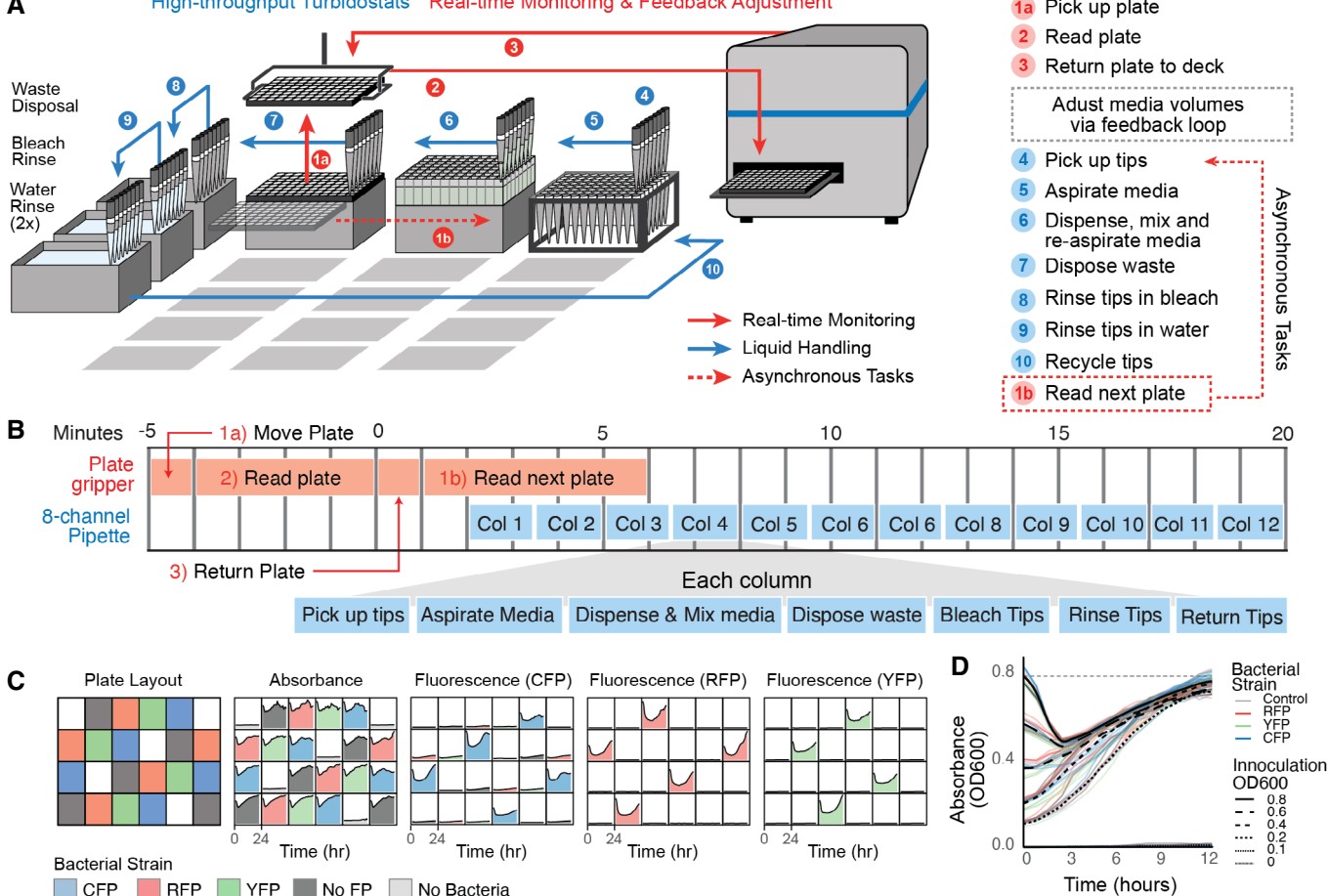

**Figure 2. High-throughput turbidostats.**

A High-throughput turbidostat summary for up to 480 simultaneous turbidostats. Bacterial populations are housed in 96-well clear-bottom plates on the deck of a liquid-handling robot. Liquid handling is used to create a turbidostat in every well, continuously refreshing each population by diluting the bacterial culture from a respective deep-well media reservoir on deck. An integrated plate reader is used to monitor absorbance, luminescence, or fluorescence readouts for each culture. Movements by robotic pipette (blue arrow) and plate reader (red arrow) are shown. Dotted lines indicate tasks that are executed asynchronously, and require 10 min per plate.

B Step-by-step summary of high-throughput turbidostat method, executed asynchronously.

C Plate layout of real-time absorbance, CFP, RFP, and YFP fluorescence readings of 96 simultaneous cultures inoculated with either no bacteria, FP-null bacteria, and CFP, RFP, or YFP-expressing bacteria. Data shown from 24 representative wells.

D Real-time absorbance measurements of 96 *E. coli* BL21 cultures inoculated at ODs of 0, 0.1, 0.2 0.4, 0.6, 0.8, which equilibrate to a set point of 0.8 within 12 h, consistent with simulation (Appendix Fig S3).

### High-throughput perturbation analysis of metabolites

We next sought to use high-throughput turbidostat tracking to address an outstanding question in metabolic engineering by systematically mapping the chemical landscape that supports bacterial growth and protein expression. To do this, we surveyed the contributions of carbon, nitrogen, and phosphorus on growth and recombinant protein production by permuting chemical gradients for these metabolites in high-throughput using our multiplexed turbidostat maintenance protocol. These dependencies, while seemingly well-studied, have not been explored in depth. Truly comprehensive mapping requires sufficient conditions, replicates, and controls, long-term maintenance of log-phase growth, and real-time monitoring, each of which is trivial to implement with Pyhamilton.

It has traditionally been thought that cells regulate protein production by allocating their resources to optimize for both expression and growth (Li *et al*, 2014; Mori *et al*, 2017). However, it has recently been shown that in either carbon-, nitrogen- or phosphorus-limiting conditions, cells are able to fine-tune their ribosomal usage to maintain equal levels of protein (Li *et al*, 2018). Thus, we wondered whether exploration of the entire metabolite landscape (Fig 3A) could more rigorously identify bacterial growth conditions optimized for recombinant protein production. To do this, we inoculated cultures with *E. coli* BL21, a strain commonly used for recombinant protein production in metabolic engineering or biomanufacturing, engineered for high constitutive expression of a fluorescent protein (CFP) (Sarabipour *et al*, 2014).

In a single experiment spanning 36 h with no user intervention, we simultaneously quantified the equilibrium log-phase growth rates and respective fluorescence levels of 300 individual turbidostats, representing 100 different media compositions in triplicate (Fig 3B). Cells were grown in modified M9 media containing 100 different ratios of carbon, nitrogen, and phosphorus and the cultures were maintained in log-phase growth for 36 h with feedback control (Supplemental methods). All cultures grew within +/− 20% of the standard M9 media growth rate, with the exception of cultures that were starved of both carbon and phosphorus (Fig 3C). We observed that increases in growth rate are primarily correlated with increases in phosphorus (independent of nitrogen or carbon levels), which is likely a result of increased DNA synthesis. Further, in phosphorus-limiting conditions, we find that the depressed growth rate can be rescued by supplementing carbon, but not nitrogen, suggesting that carbon precursors are a more limiting reagent than amino acids in metabolism (Fig 3C). Consistent with previously published results (Li *et al*, 2018), we observe that the total amount of protein is generally not affected by limiting carbon or nitrogen, nor by supplementing the cells with excess of either nutrient. However, we additionally find that when phosphorus is limited (0.25X), excess carbon supplementation not only rescues the growth rate of the culture (Fig 3C, Dataset EV2), but also results in an increase in total fluorescence (Fig 3D, Dataset EV2). Since we observe negligible growth defects, this finding suggests that on a per-cell basis, supplementing carbon in phosphorus-limiting conditions (such as in the soil (Ostertag, 2008; Vitousek *et al*, 2010) or P-limited lakes (Hessen, 1992)) can shunt bacterial metabolism

from DNA/mRNA synthesis to protein translation without sacrificing growth. Collectively, these findings demonstrate that Pyhamilton enables researchers to answer rigorous metabolic engineering questions by enabling facile, low-consumable, yet rich hypothesis-generating experiments.

## Discussion

Liquid-handling robots have traditionally automated workflows that were explicitly designed for human researchers rather than enabling new high-throughput experimental modalities. Pyhamilton is an open-source Python framework intended for experiments that could never be done by hand, such as protocols that must pipette continuously for multiple days, perform complex calculations about future steps based on real-time data, or make use of hardware that is more sophisticated than any hand-held multichannel pipette.

We showcase these improved capabilities by simultaneously quantifying the metabolic fitness landscape of 100 different bacterial growth conditions to identify ideal conditions for recombinant protein production. Though recent fluidic advances have enabled the maintenance of many continuous cultures (Gupta *et al*, 2017; Wong *et al*, 2018; Haby *et al*, 2019), our liquid-handling platform can accommodate several times as many. Moreover, liquid-handling systems can easily incorporate a plate reader for real-time reporter monitoring, which vastly expands the types of questions that can be approached with facile, multiplex solutions. For example, one could maintain cultures of, and accurately quantify any reporter output for massively-parallel experiments including genetic knockout or CRISPR collections (Baba *et al*, 2006; Peters *et al*, 2016), mutagenesis variants (Miyazaki & Takenouchi, 2002), or even small-molecule compound libraries (Geysen *et al*, 2003). With high accuracy, any suspension culture of mixed populations could be maintained in log-phase growth for days in order to study transient invaders into microbial communities (Amor *et al*, 2020) or even microbiome system dynamics (Lloyd-Price *et al*, 2017). The advent of small-molecule fluorescent reporters for metabolic fitness (Zhao & Yang, 2015), pH (Zhang *et al*, 2016; Si *et al*, 2016), and $CO_2$ (Zhujun & Seitz, 1984), in addition to the hundreds of fluorescent protein sensors available to the synthetic biology community at large (Palmer *et al*, 2011; Hu *et al*, 2018), underscore the many potential applications of being able to multiplex and quantify changes in growth, gene expression, and the environment in real-time.

Presently, Pyhamilton is only extensible to Hamilton robots. However, since it uses a platform-independent, web-based protocol (HTTP) and common readable data format (JSON) to bridge Python and the Hamilton Scripting Language (HSL) (Appendix Fig S1), Pyhamilton could be ported to other biological automation systems that provide an API, such as Tecan or alternative platforms.

As such, Pyhamilton is a small part of an ongoing transition to a paradigm which leverages insights from computer science (Bär *et al*, 2012) and applies them to biology. Similar to how Bioconductor (Gentleman *et al*, 2004) and The Biopython project (Cock *et al*, 2009) have revolutionized computational biology,

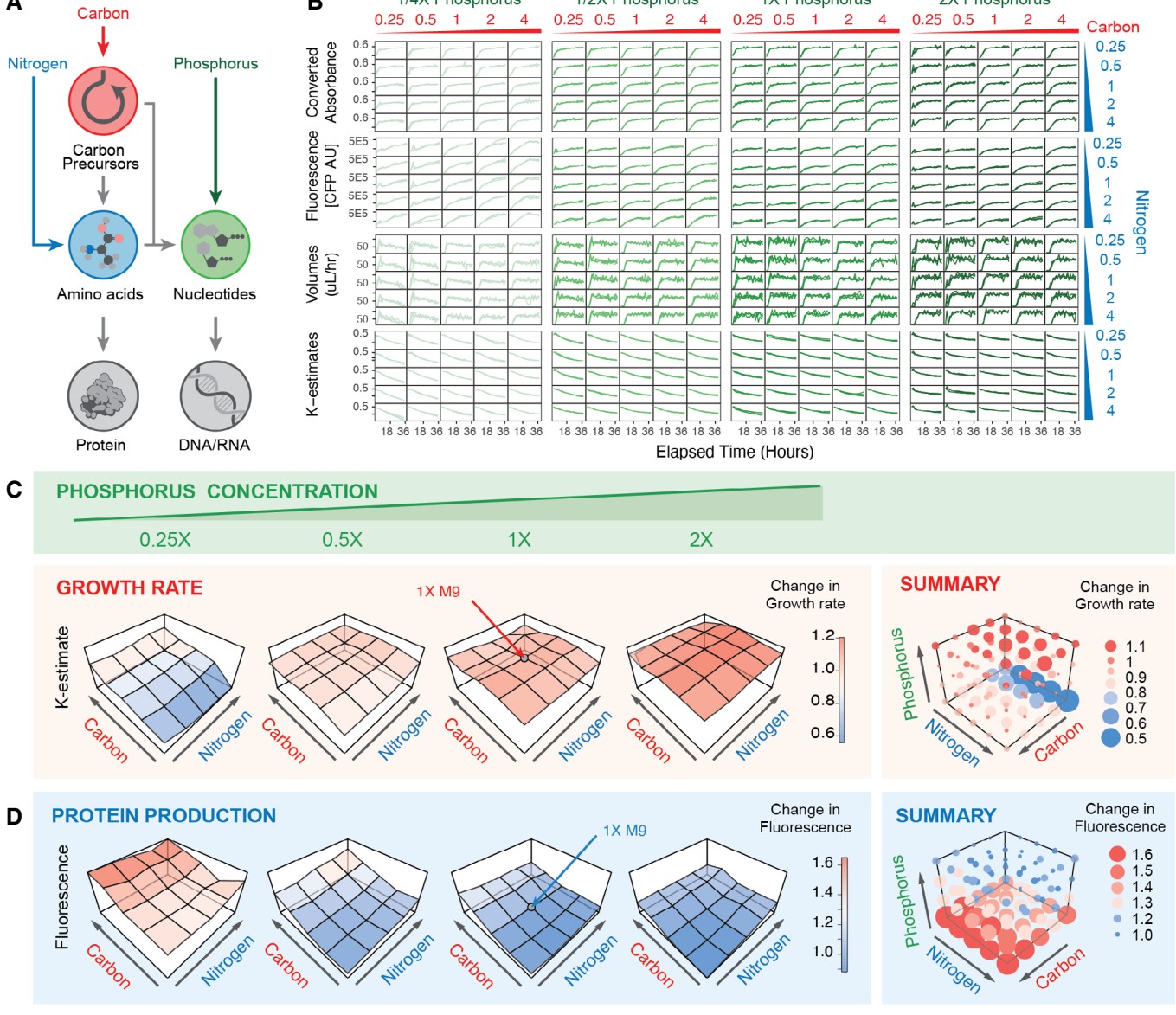

**Figure 3. Metabolic profiling of protein production.**

A Schematic flow of carbon, nitrogen and phosphorus nutrients in protein and nucleotide production.

B Real-time absorbance and fluorescent reporter monitoring for BL21 *E. coli* expressing CFP in 100 various M9 media compositions (*n* = 3 per condition). Real-time calculations of volumes/h and estimates for k-value convergence shown.

C (left) Average growth rate for each media composition plotted as a 2-dimensional fitness landscape of carbon and nitrogen, for four concentrations of phosphorus. (right) Summary of all 100 conditions shown as 3D fitness landscape colored by growth rate (blue = low, red = high). Size of dot indicates absolute deviation from average 1X M9 media composition.

D (left) Average amount of protein expression (measured by fluorescence) of each media composition plotted as a 2-dimensional fitness landscape of carbon and nitrogen, for four concentrations of phosphorus (right). 3D protein production landscape of all 100 conditions colored by amount of fluorescence (blue = low, red = high). Size of dot indicates absolute deviation from average 1× M9 media composition.

bioinformatics, and genomics, our hope is that by making this software open-source and freely available, a community of scientists and developers could begin to similarly transform bioautomation. The experiments we have described represent only a small sampling of many possible Pyhamilton applications. Collectively, they highlight the potential of high-throughput robotic systems to transcend the repetitive processes for which they were conceived and directly address broad questions in microbiology, genetics, and evolution that are beyond the physical capabilities of human researchers.

# Materials and Methods

## Reagents and Tools table

| Reagent/Resource | Reference or source | Identifier or catalog number |
| --- | --- | --- |
| **Experimental models** | | |
| BL21(DE3) | New England Biolabs | Cat #C2527I |
| S2060 | Addgene | Cat #105064 |
| **Recombinant DNA** | | |
| pRSET-B YFP | Addgene | Cat #108856 |
| pRSET-B mCherry | Addgene | Cat #108857 |
| pRSET-B CFP | Addgene | Cat #108858 |
| **Chemicals, enzymes and other reagents** | | |
| Carbenicillin | Gold Biotechnology | Cat #C-103-5 |
| Chloramphenicol | Gold Biotechnology | Cat #C-105-5 |
| **Software** | | |
| *Pyhamilton* | https://github.com/dgretton/pyhamilton | |
| Hamilton Run Control Software | | |
| **Other** | | |
| Hamilton STARlet | Hamilton Company | Cat #173020 |
| 1,000 µl Pipetting Channels, 8 channels | Hamilton Company | Cat #173081 |
| CO-RE 96 channel Multi Probe Head | Hamilton Company | Cat #199090 |
| iSWAP Plate Handler | Hamilton Company | Cat #190220 |
| HEPA Flow Hood, UV | Hamilton Company | Cat #55502-01 |
| CLARIOstar Multi-Mode Microplate Reader | BMG LABTECH | Cat #0430-101 |

## Methods and Protocols

### Robotic equipment set-up and interfacing

A Hamilton Microlab STARlet 8-channel base model was augmented with a Hamilton CO-RE 96 Probe Head and a Hamilton iSWAP Robotic Transport Arm. Air filtration was provided by an overhead HEPA filter fan module integrated into the robot enclosure. A BMG CLARIOstar luminescence multimode microplate reader was positioned inside the enclosure, within reach of the transport arm. *Software.* A general-purpose driver method was created using MicroLab STAR VENUS ONE software and compiled to Hamilton Scripting Language (HSL) format. Instantiation of this method and management of its local network connection was handled in Python. A new Pyhamilton-compatible supporting Python package provided an overlying control layer interface to the CLARIOstar plate reader. We used Git to develop and version control the packages and the specific Python methods used for each experiment; our software implementation can be found on github at: https://github.com/dgretton/pyhamilton.

### Bacterial assays

For bacterial assay validation, bacterial plaque assays were used to confirm dilutions and agar solidification. Briefly, overnight cultures of S2060 cells (Addgene bacterial strain #105064) were grown in 2XYT media (Digest Peptone 16 g/l, Yeast extract: 10 g/l, Sodium Chloride: 5 g/l; Research Products International #X15600)

supplemented with maintenance antibiotics were diluted 1,000-fold into fresh media with maintenance antibiotics and grown at 37°C with shaking at 230 rpm to $OD_{600}$ ~0.6–0.8 before use. M13 bacteriophage were serially diluted 100-fold (4 dilutions total) in $H_2O$. 20 µl of bacterial were added to 100 µl of each phage dilution, and to this 200 µl of liquid (70°C) "soft" agar (2XYT media + 0.6% agar) supplemented with 2% Bluo-Gal was added onto a well of a 24-well plate already containing 235 µl of hard agar per well (2XYT media + 1.5% agar, no antibiotics). To prevent premature cooling of soft agar, the soft agar was placed on the robot deck in a 70°C heat block. After solidification of the top agar, plates were incubated at 37°C for 16–18 h. Source code from our implementation can be found at: https://github.com/dgretton/roboplaque

### Population dynamics experiments

Briefly, 96-well clear-bottom plates were filled with 100 µl of water in each well. Point-spread analysis was initiated by adding colored dye to the first well, and liquid transfers were compiled and executed in real time using Pyhamilton on a Hamilton Microlab STARlet. Source code for our implementation can be found at: https://github.com/dgretton/pyhamilton_population_dynamics

### Feedback controller algorithm

Bacteria optical density (OD) was modeled to evolve as: $x = x_0 e^{kt}$, where $x$ is the culture OD, $x_0$ is the initial OD, $k$ is the bacteria

exponential growth constant ($k$-value) in reciprocal hours, and $t$ is elapsed time in hours. A media replacement cycle is modeled as dilution of a culture by instant uniform mixing with transparent media of a fraction $y$ of its initial volume, which linearly scales its OD $x$ to a new OD $x'$ (e.g. if a 100 µl culture is at OD 0.3 and $y = \frac{1}{2}$, then the replacement is modeled as diluting with 50 µl transparent media, and the final OD $x'$ is 0.2), summarized as:

$$x' = \frac{1}{1+y}x.$$

The culture OD is to be maintained at a constant setpoint, $x^{set}$. In each cycle $i = 0,1,2,\ldots$, each representing a time interval $\Delta t$, the turbidostat controller is responsible for producing an output command and state update according to a transfer function:

$$(y_i, \phi_i) = f(x_i, \phi_{i-1}).$$

where $y_i$ is the new controller output command as a fraction of the turbidostat volume, $\phi_i$ is the new controller internal state, $x_i$ is the present OD measurement, and $f(x_i, \phi_{i-1})$ is the controller transfer function based on the OD measurement and the previous controller state $\phi_{i-1}$. The controller state may depend on the history of prior OD measurements $x_0,\ldots,x_{i-1}$ and prior controller commands $y_0,\ldots,y_{i-1}$.

### Specific controller state

A feedback controller with a distinct state was created for each culture. The controller state is a triple $\phi_i = (k_i^e, y_i)$: the present OD measurement, $x_i$; the current estimate of the culture's growth k-value, $k_i^e$; and the output command, $y_i$.

### Transfer function

The transfer function updates the three state variables and computes an output. From the model equations, the current k-value, given a new measurement $x_i$ taken an interval $\Delta t$ after the previous replacement executed, is

$$k_i = \frac{\ln\left(\frac{x_i}{x_{i-1}}(y_{i-1}+1)\right)}{\Delta t}$$

This $k_i$ contributes to the state $k$-value estimate $k_i^e$ through a first-order linear filter to dampen the effect of measurement noise. The output to restore the turbidostat OD to the setpoint is

$$y_i = \max\left(0, \min\left(y^{max}, \frac{x_i e^{k_i^e \Delta t}}{x^{set} - 1}\right)\right).$$

where the final output $y_i$ is subject to physical limits, being both nonnegative and not greater than the largest volume the robot can move with a pipette tip as a fraction of the turbidostat volume, appearing as $y^{max}$. After output limiting, $y_i$ is saved in the controller state. Controller was developed as an abstract Python class and tested in simulation with mechanical and measurement noise models before application in experiments (Appendix Figs S3 and S6). Filtered k-value estimates were used to draw conclusions about bacterial growth rates. Source code for implementation can be found at: https://github.com/dgretton/many_basic_turbidostats/blob/master/turb_control.py.

### On-deck turbidostat cultures
#### Peristaltic pump array

To pump media onto the deck, up to seven miniature 12 volt, 60 ml/min peristaltic pumps ("fish tank pumps") were actuated by custom motor drivers. A Raspberry Pi mini single-board computer received instructions over local IP and commanded the motor drivers via I²C (extended pump configuration details, see DeBenedictis *et al*, 2020). Between each iteration, the reservoir was filled with fresh 2XYT media, and then media was added to each bacterial turbidostat growing in a 24-well plate, based on OD and parameter estimation. Each turbidostat was then sampled by aspirating culture into a 96-well plate reader plate, which was then read using an integrated ClarioStar plate reader. Excess media was then drained from the reservoir, and all other system components were rinsed 1× 5% bleach and 4× water between each iteration. S2060 bacterial strains were grown in 2XYT media supplemented with antibiotics. Source code for implementation can be found at: https://github.com/dgretton/many_basic_turbidostats.

### High-throughput turbidostat cultures
#### Cell strains and growth conditions

To generate fluorescent reporter strains, plasmids pRSET-B YFP, pRSET-B mCherry, and pRSET-B mCherry were transformed into *E. coli* strain BL21(DE3) (New England Biolabs). Plasmids were a gift from Kalina Hristova (Addgene #108856, Addgene #108857, Addgene #108858). Bacteria cells were grown overnight in LB media, and then conditioned to grow in M9 Minimal Media: 33.7 mM $Na_2HPO_4$, 22.0 mM $KH_3PO_4$, 8.5 mM NaCl, 9.35 mM $NH_4Cl$, 0.4% Glucose, 1 mM $MgSO_4$, 0.3 mM $CaCl_2$, 1 µg biotin, 1 µg thiamin, 1× trace elements (Trace elements solutions (100× stock solution, 100 mg/l $MnCl_2.4H_2O$, 170 mg/l $ZnCl_2$, 43 mg/l $CuCl_2.2H_2O$, 60 mg/l $CoCl_2.6H_2O$, 60 mg/l $Na_2MoO_4.2H_2O$). For modified M9 Media, Phosphorus, Carbon, and Nitrogen sources were increased or decreased by 2 or 4 fold. For turbidostat inoculations, starter cultures were grown overnight at 37°C for 16–18 h, and then diluted 1:100, and then grown for another 4–8 h until in log-phase growth. When each strain reached log-phase growth (OD 0.6–0.8), cultures were first diluted to an OD of 0.6 and then turbidostats were inoculated 1:100 into 175 µl in 96-well plate reader plates (Black/Clear flat bottom polystyrene plates, Corning #3631) prior to initiation of the robotic method. Unlike the first on-deck turbidostat culture method, in which media was pumped onto the robotic deck, for high-throughput tubidostates, 2 ml of media for each well was aliquoted into a 96-deep well plate (Thomas Scientific, Item #1149J23). Media is replenished daily, or when running low. The robot deck was organized as described in Fig 2A.

#### Antibiotics

Antibiotics (Gold Biotechnology) were used at the following working concentrations: carbenicillin, 50 µg/ml; chloramphenicol, 40 µg/ml. Source code for implementation can be found at: https://github.com/dgretton/many_asynchronous_turbidostats.

## Data availability

Source code can be found at: https://github.com/dgretton/pyhamilton. Data from the metabolic fitness landscape experiment can be found in Dataset EV2.

Expanded View for this article is available online.

## Acknowledgements

We are grateful to Alvaro Cuevas of Hamilton Robotics for his examples, guidance, and assistance in making use of the Original Equipment Manufacturer (OEM) interface, along with the rest of Hamilton Robotics. We thank Jason Yang, Stephen Von Stetina, Ethan Alley, Brian Wang, Samantha Shepherd, Timothy Erps, and the three reviewers for thoughtful comments and discussion. EAD was supported by the National Institute for Allergy and Infectious Diseases (F31 AI145181-01). EJC was supported by the Ruth L. Kirschstein NRSA fellowship from the National Cancer Institute (F32 CA247274-01). This work was supported by the MIT Media Lab, an Alfred P. Sloan Research Fellowship (to KME), gifts from the Open Philanthropy Project and the Reid Hoffman Foundation (to K. M. E.), the National Institute of Digestive and Kidney Diseases (R00 DK102669-01 to KME), the Burroughs Wellcome Fund (IRSA 1016432 to KME) and the DARPA Safe Genes Program (N66001-17-2-4054 to KME). The findings, views, and/or opinions expressed are those of the authors and should not be interpreted as representing the official views or policies of the Department of Defense or the U. S. Government.

## Author contributions

*Software*: DWG. *Conceptualization*: EJC, DWG, EAD, and KME. *Methodology*: EJC, DWG, EAD. *Validation*: EJC, DWG, EAD. *Formal Analysis*: EJC. Investigation: EJC, DWG. Writing – Original Draft: EJC. Writing – Review & Editing: EJC, DWG, EAD, and KME. *Visualization*: EJC. *Funding Acquisition*: KME.

## Conflict of interest

The authors declare that they have no conflict of interest.

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
