## [Review Process File · Molecular Systems Biology]

Enabling high-throughput biology with flexible open-source automation

Emma Chory, Dana Gretton, Erika DeBenedictis, and Kevin Esvelt

DOI: [10.15252/msb.20209942](https://doi.org/10.15252/msb.20209942)

Corresponding author(s): Kevin Esvelt (esvelt@mit.edu)

Review Timeline:

Submission Date:	22nd Aug 20
Editorial Decision:	30th Oct 20
Revision Received:	29th Nov 20
Editorial Decision:	13th Jan 21
Revision Received:	3rd Feb 21
Accepted:	5th Feb 21

Editor: Maria Polychronidou

Transaction Report:

Thank you again for submitting your work to Molecular Systems Biology. I apologize for the unusual delay in getting back to you with a decision, which was due to the late arrival of the referee reports. We have now heard back from the three referees who agreed to evaluate your study. Overall, the reviewers acknowledge that the proposed platform seems potentially interesting. However, they raise a series of concerns, which we would ask you to address in a revision.

I think that the issues raised by the referees are clear and relatively straightforward to address, and I therefore see no need to repeat the points listed below. One important point raised, refers to the need to discuss how generalisable the platform is and what is the extent of its dependency on Hamilton products. Please let me know in case you would like to discuss in further detail any of the issues raised. All issues raised by the referees would need to be satisfactorily addressed.

On a more editorial level, we would ask you to address the following.

REFEREE REPORTS

Reviewer #1:

This paper presents a software library (Pyhamilton) that allows Hamilton robots to be programmed at a higher level of abstraction as compared to using the native software.

High level comments:

- In the introduction, another aspect of automation of this type is to allow protocols to be easily shared, modified, etc. That point should be made.
- An example given of the types of improvements available with this software is that pipetting liquids over two 24 well plates is possible. It would be nice if for every type of improvement possible a "class categorization" is created and a table is presented with each type of categorization. For

example: multi-plate pipetting, liquid transfer calculations, feedback control, etc. For each category the benefits could be listed (redundancy, throughput, re-use, cost, etc.). This would be a quick visual way to see what types of things are possible and why someone would employ these techniques.

- For each of the experimental examples shown, it would be nice to see the cost, time, and personnel requirements to do this manually and with the Hamilton Venus software as points of comparison. Both of these might be back of the envelope but having some idea of the order of magnitude savings would be nice.

- What makes this only for Hamilton robots? Could this be ported to other robots? What layers of abstractions and backends would allow this to be ported to other systems? How generalizable are the concepts? Will this work for future Hamilton robots?

Case study comments:

- Pipetting flexibility - my assumption is that the accuracy is maintained. Also I assume there are other configurations that can be explored like which tips actually dispense, the volume per pipette, etc. This would be good to understand. It would also be good to understand how hardware limitations are reflected in the software. Are they hard constraints? Can they be changed if more flexible hardware comes out? How are they checked? What feedback is given to the user when running the code if HW capabilities are violated?

- Liquid transfer calculations - not a lot of feedback here. I guess the real savings is how hard this is with the native software and if experimental needs of this type change enough that doing it quickly in Pyhamilton represents a real savings. That being said, I think this is well motivated and a good example of the power of the approach.

- Feedback control - The figure associated with example this would benefit from seeing what the alternative approach would look like. Again, I believe that Pyhamilton can do this. I think many readers will wonder how much easier it is or better. Personally I think feedback would be the most interesting to watch how it can be used to tune protocols across robots so that they all get similar results with labware and reagents from different vendors. Essentially how the robots "self-tune" to different operating conditions and environments. Some discussion of this possibility would be nice.

- Turbidostats - here the general classification of asynchronous feedback and decisions as a general class of operations would be powerful. A good example of a couple concepts in one example. The extent that this excites readers depends on their ability to abstract away the application into the programming concepts presented (which I have found to be a challenge for many experimental biologists).

- Metabolites - I lost track of the key element that made Pyhamilton vital to this work other than the fact that a liquid handling robot would be needed for this task.

Overall Comments:

- This is a well written paper

- The figures are very high quality

- This is a key advance for a very specific area of bio-design automation that desperately needs more contributions (and visible publications)

- The experimental results are provided for experimentalists (clearly). That being said, they are largely irrelevant to a programming language. A programming language will enable a set of operations in HW. What someone does with those operations is up to them. The programming language needs to be efficient, sufficiently abstract, expressive, and portable. None of that is the focus of the paper. I totally understand the challenge the authors have in this space trying to get this into a venue read by biologists and not computer scientists. It would be nice to see a more programming oriented white paper or something similar made available to the python programming community that complements this work.

Reviewer #2:

In this manuscript the authors successfully expanded the experimental capabilities of robotic liquid handlers by developing a new, open-source, flexible Python software platform for Hamilton STAR, STARlet and vintage robots. The package, Pyhamilton, makes programming and running advanced methods on Hamilton robots much more straightforward, and also enables physical robot motions not allowed in the manufacturer software. Such programming enables more complex experiments, which the authors demonstrate with a growth screen of E. coli BL21 under a large number of different carbon, phosphorus, and nitrogen levels.

Although the developed package solves an existing problem with automating experimental assays in Hamilton platforms, a major limitation is that it appears to be limited to only Hamilton products. As such, there does not appear to be any broadly applicable new capabilities which can be ported to other liquid handling platforms to increase the impact of the code base and demonstrated methodologies. In line with this, it brings to question the level of development of Hamilton's software and if there are similar limitations from other lab automation hardware providers (such an evaluation was not provided).

Major comments

1. Given that the overall work scheme is targeting the Hamilton robotic platforms only, it is suggested to change the current paper's title to imply the same. There is no mention of applicability of the work to other liquid handling / automated platforms. For example, line 90 states that there is a major limitation of the existing software which makes it no faster than hand-pipetting. Is this true of similar platforms to Hamilton's products?

2. The authors demonstrated that they were able pipetting liquids over two 24-well plates simultaneously and thus enabling the robot's full mechanical capability. Can the same procedure be applied to 96-well plates as well and other standard plate formats? Please address the scope of applications that this new capability addresses. Along these lines, can the authors comment on if it is possible to program in any motion that is physically possible on the Hamilton robot with the code base? Providing more examples or a list of expanded functionalities would be ideal, if possible. Line 339 hints at an expansive list.

3. Line 180. There is a general lack of details for the described turbidostat functionality and how pyHamilton makes this functionality uniquely possible. How were the bacterial turbidostats "maintained"? Specifically, it seems warranted to describe the cultivation conditions in more detail in the main text such as mixing and aeration control, volumes, the range of interventions (adding media, sampling from a given well) possible with the number of replicates, etc. Denoting the specific actions that are impossible with the available manufacturers code vs. pyhamilton seems to be an opportunity to showcase the utility of the code base. Again, is this functionality usable with other liquid handling platforms?

4. It was mentioned several times that Pyhamilton makes programming more complex robotic methods much easier than in the manufacturer's software, and enables the programming of methods that are impossible in the manufacturer's software (such as the feedback loop used to control the OD in the turbidostat method). It would benefit readers who are not familiar with Hamilton products (which is probably the vast majority of the readership) to include a brief description of why this is so. This description could be made stronger by including an example script of the Hamilton control software and what the limitations are for the closest best option without pyhamilton (if this is allowable by their terms of service).

5. At the Github pyHamilton's repository authors mentioned "... on Windows XP and Windows 7". What is required to run the software implementation on the most recent Windows and their dependencies, as these versions of Windows are obsolete and unusable on most institutional networks?

6. It is unclear what extent of human intervention is necessary when running the robots using pyHamilton for the presented test cases? These should be included in each example to provide context. Initial set up and interventions during run time.

7. For designing/testing and running experiments, is it possible to simulate a given design/experiment before each run? I.e., is there a simulation mode or if not, why not?

8. Section (High-throughput perturbation analysis of metabolites), it can be assumed that the authors used strains with more or less similar growth rates. Changing media components to optimise expression and growth is not challenging using the system presented, however screening strains that differ in their growth rates would be challenging if such screens were run at the same time. Please address the applicability of the current code base to such a setup of using strains with different growth rates.

Minor comments

1. The cross-contamination tests were run for 24 hours. This seems like a rather limited time duration and should be put in that context.

2. A lot of interesting methods were developed for the specified robotic platform. It would be of great help in order to replicate the work for other researchers to include small video clips to each developed method, if possible.

3. Line 545: It isn't entirely clear from this description how Pyhamilton interacts with the Hamilton liquid handling robots. Is the manufacturer's software running at the same time, or does this software interact directly with the robot?

4. What was the trace elements recipe, line 665?

5. Fig. 1A/C/G/J: Is this font required? If not, it would be beneficial to change it to something easier to

read.

6. Fig. 1A: The references to Fig 1C, 1F, 1H should probably read 1C, 1G, 1J

7. Missing axis label for figure 2d, please add.

8. Short description of the robotic method, corning (line 673) Item#3631, is required.

9. Line 255: Should "All cultures grew within +/- 20% of M9 media growth rate" read "20% of the standard M9 media growth rate"?

10. Fig 3C: You should have the phosphorus concentration levels labelled as in 3B, since the line implies a continuous gradient of phosphorus and gives less information on the concentration differences between the plots.

11. Can you please provide a reference for 2XYT medium used, Line 558?

Here is a list of grammatically incorrect and misspelling statements that needs a careful review beside a thorough edit for more clarity and grammar check:

1. *E. coli* should be italic or underlined throughout the text.

2. Typo in media medium for singular, line 674.

3. Line 88: Comma needed after simultaneously

4. Line 135: shouldn't read "the python"

5. Fig. 1B caption: allow for improve throughput of laboratory across 24-well plates should be rephrased

6. Line 241: The paragraph break at the end of this sentence should be reconsidered, as the description of the experimental setup is split between the paragraphs.

7. Line 563: Bacterial should read bacteria

8. Text under figure 1, the following sentence needs rephrasing? "(C) ... bacteriophage plaque assay show".

9. Figure 2, missing 1a?

10. For cultures of S2060 (Line 558), Add the Addgene reference number. Is this (Bacterial strain #105064) correct?

Reviewer #3:

The authors develop an open-source Python platform with the goal of improving the flexibility of Hamilton liquid handling systems and enabling high-throughput experiments of greater complexity. Their system allows for pipetting steps to be coupled in real time to read outs from a standard microplate reader, enabling the setup of controllers that dynamically respond to the current state of the experiment. The authors apply their platform to a number of different test cases centered around cell growth. Their main examples focus on functionally converting multiple well-plates into turbidostats with cultures that are maintained at a fixed OD. They further utilize this capability to study the impact of carbon, nitrogen, and phosphorus sources on both cell growth and protein production in defined media. Additionally, they develop and implement a cleaning process that allows for the re-use of pipette tips.

This work represents a significant improvement to the capabilities of standard tip-based liquid handling systems. The implementation of controller modules that can continuously modify reaction conditions in a well-plate is particularly exciting, and the example of converting standard plates into turbidostats is both convincing and impactful. Their systematic interrogation of the impacts of

limiting nutrients on growth rate and protein expression yields highlights the flexibility of the system and provides insight into the metabolic processes of bacteria under resource limited growth conditions. The manuscript is well written, and the examples are presented in a clear and concise manner, with one exception noted below.

Major comments

- The purpose and meaning the population dynamics simulation in figure 1 d-g is not clear. Additionally, the treatment of the topic in the text is cursory. It seems that the purpose is to show the ability to pipette different quantities of liquid in complex and programmable patterns. This is obscured by the example of population dynamics given the instrument is pipetting dye. A more detailed discussion of how this relates to experimentally testing population dynamics is warranted.
- The data associated with the cleaning protocol in figure 2 c only show that no cross contamination between wells occurred, not that the media source wells were not contaminated. This should also be proven.

Minor comments

- The limitations of the system are not discussed in detail.
 - o In the case of the plate-based turbidostats what are the limitations on the growth rate of the organism that can be maintained, both within a single experiment and alone? What are the maximum number of cultures that could be maintained in a given experiment? Is this a limitation of the footprint available for the pipetting system, or a limitation of the transfer speed?
 - o One of the major limitations of tip-based robotic systems is the long setup times for complex transfer protocols. What is the timescale for performing several hundred transfers, as was done in the population dynamics example, and is this significant on the timescale of the experiment.

Reviewer #1:

Summary: This paper presents a software library (Pyhamilton) that allows Hamilton robots to be programmed at a higher level of abstraction as compared to using the native software.

High level comments:

- In the introduction, another aspect of automation of this type is to allow protocols to be easily shared, modified, etc. That point should be made.

We have modified the following sentence to highlight that protocols can be easily shared and modified:

“To enable flexible high-throughput experimentation, we developed Pyhamilton, a Python package that facilitates high-throughput operations within the laboratory, with protocols that can be easily shared and modified.”

We have also reiterated this point in the methods section and Supplemental Video 1, with additional descriptions on how the protocols can be obtained through github.

- An example given of the types of improvements available with this software is that pipetting liquids over two 24 well plates is possible. It would be nice if for every type of improvement possible a "class categorization" is created and a table is presented with each type of categorization. For example: multi-plate pipetting, liquid transfer calculations, feedback control, etc. For each category the benefits could be listed (redundancy, throughput, re-use, cost, etc.). This would be a quick visual way to see what types of things are possible and why someone would employ these techniques.

While we would like to provide a white paper on Pyhamilton as suggested by the Reviewer, for the moment we have included an updated README file as well as a supplemental document in standard Python Documentation format which thoroughly details the class categorizations used in each method with substantially more detail than we could provide in the manuscript alone. This documentation is also available at <https://dgretton.github.io/pyhamilton-docs/>. We thank the reviewer for insisting on this, as we agree that it is critical for proper sharing and dissemination--- our primary motivation behind the development of Pyhamilton in the first place. Supplemental Table EV2 also now contains cost, personnel time, and re-usability of all consumables employed in each of the methods we describe.

- For each of the experimental examples shown, it would be nice to see the cost, time, and personnel requirements to do this manually and with the Hamilton Venus software as points of comparison. Both of these might be back of the envelope but having some idea of the order of magnitude savings would be nice.

We thank the reviewer for this excellent suggestion. As described above, we have added a supplemental table (Table EV2), which contains the relevant consumables, robotic equipment, suggested manufacturer and part number, estimated cost per item/experiment, information on personnel time, iteration time, githubs links to each example script and robotic layout file. We have additionally included links to supplemental videos of each method and a respective recording of a simulation run.

- What makes this only for Hamilton robots? Could this be ported to other robots? What layers of abstractions and backends would allow this to be ported to other systems? How generalizable are the concepts? Will this work for future Hamilton robots?

The asynchronous call-and-response command architecture is extensible to all robot platforms. The Hamilton-specificity is due to the particular command set we developed, which makes reference to Hamilton-specific parameters (e.g. an integer that indicates which dispense mode to use). These specific commands are also encoded in the Pyhamilton interpreter, on the robot side. Pyhamilton itself is agnostic to any particular command set. Additional custom commands can be added without changing the Python package code, so Pyhamilton will easily support new features offered by Hamilton in the future, and can be flexible outside of the Hamilton ecosystem as well. Since it uses a platform-independent, web-based protocol (HTTP) and common readable data format (JSON) to bridge Python and HSL, Pyhamilton can be ported to other platforms by writing an interpreter for the new platform that can send and receive network GET and POST requests, a ubiquitous feature set. The same basic concepts of aspiration, dispensing, mixing, moving plates, etc. are universal across all robotics that interact with standard biological labware. Since pyhamilton offers commands at this level of abstraction with a simple coupling, it is well suited to adapt to new platforms (and its name may need to be revised in the future). To address this in the text, we have added a new supplemental figure (Supp. Fig 1) to summarize the layers of abstraction and added the following text to the discussion section:

“Presently, Pyhamilton is only extensible to Hamilton robots. However, since it uses a platform-independent, web-based protocol (HTTP) and common readable data format (JSON) to bridge Python and the Hamilton Scripting Language (HSL) (Supp. Figure 1), Pyhamilton could be ported to other biological automation systems that provide an API, such as Tecan or alternative platforms.”

Case study comments:

- Pipetting flexibility - my assumption is that the accuracy is maintained. Also I assume there are other configurations that can be explored like which tips actually dispense, the volume per pipette, etc. This would be good to understand. It would also be good to understand how hardware limitations are reflected in the software. Are they hard constraints? Can they be changed if more flexible hardware comes out? How are they checked? What feedback is given to the user when running the code if HW capabilities are violated?

At the moment, the channels used to execute specific pipetting actions, and the volumes used in each, must be given exactly by scripts utilizing Pyhamilton. This choice is arbitrary, and permutations are possible. Except in rare cases at the far extremes of the robot's motion, any channel permutation will always work just as well as any other. The only difference is in how long the Hamilton STARlet might take to finish the step; for example, if the channels are specified in reverse, the Hamilton (not Pyhamilton) will automatically pipette one channel at a time, because no two channels can operate simultaneously due to the order constraint. Because Pyhamilton adheres to python convention, channels specified as ranges of integers should “just work” optimally, so such inefficiencies are unlikely. The only constraints imposed by Pyhamilton regard data types and required parameters for each command. If new hardware requiring different data (e.g. adding more channels) or totally new data types becomes available, commands can be redefined or added, and particularly useful commands can be made permanent defaults in the open source code. While Pyhamilton checks for well-formed commands before they are sent and provides detailed and readable errors about missing or extra parameters, it is limited in that it cannot report errors that Hamilton's core software does not detect. These include the “Position Errors” raised when a pipette or gripper cannot reach the specified labware, which are unknown before runtime (and not even detectable by Hamilton's simulator software). In cases like these, Pyhamilton raises the most specific possible

errors, such that debugging can proceed more smoothly in the last stages of method development when working live with the robot.

- Liquid transfer calculations - not a lot of feedback here. I guess the real savings is how hard this is with the native software and if experimental needs of this type change enough that doing it quickly in Pyhamilton represents a real savings. That being said, I think this is well motivated and a good example of the power of the approach.

We thank the reviewer for this comment.

- Feedback control - The figure associated with example this would benefit from seeing what the alternative approach would look like. Again, I believe that Pyhamilton can do this. I think many readers will wonder how much easier it is or better. Personally I think feedback would be the most interesting to watch how it can be used to tune protocols across robots so that they all get similar results with labware and reagents from different vendors. Essentially how the robots "self-tune" to different operating conditions and environments. Some discussion of this possibility would be nice.

We have added several Supplemental Figures and videos which we believe clarify these possibilities. As it is quite challenging to program the feedback module in HSL (and the inability to so easily served as part of the motivation for Pyhamilton development), we have included a side-by-side video which compares programming a simple task with Pyhamilton (transferring tips between locations) to the same task using the Hamilton GUI. Since most labware is standard-issued by Hamilton, we addressed the limitations of the feedback control method by running experiments spanning many days with many different growth conditions (Supplemental Fig 5), and included simulations of how the feedback controller responds when the method is pushed to its limits (max growth rate/max number of turbidostats/etc.). We hope this addresses the reviewer's comment.

- Turbidostats - here the general classification of asynchronous feedback and decisions as a general class of operations would be powerful. A good example of a couple concepts in one example. The extent that this excites readers depends on their ability to abstract away the application into the programming concepts presented (which I have found to be a challenge for many experimental biologists).

We hope that the extended documentation, video recordings of the robot, simulation-mode videos, and side-by-side of the Hamilton GUI and Python coding will more easily distill the abstraction of programming concepts to experimental biologists. We agree that this is a broader challenge to bridge between experimentalists and programmers, and thank the reviewer for acknowledging this!

- Metabolites - I lost track of the key element that made Pyhamilton vital to this work other than the fact that a liquid handling robot would be needed for this task.

This method would not have been feasible in the absence of asynchronous method execution due to the large number of turbidostats needed to accomplish the appropriate number of replicates and experimental conditions, in addition to reading both absorbance and fluorescence on a plate reader. We have added a phrase to the introduction of the section noting that multiplexed turbidostat maintenance was required, and we have additionally added a new Figure 3b and added Supplementary Figure 6, which details the timescales required for each executable step within the method and the timescales available to maintain large number of cultures at various growth rates. We hope that for those experimentalists who are comfortable with the challenges of timing bacterial doubling times, that the visual in 3b will clarify the necessity of asynchronous timing for very high numbers of turbidostats.

Overall Comments:

- This is a well written paper
We thank the reviewer for this very kind statement.
 - The figures are very high quality
We thank the reviewer for this generous comment.
 - This is a key advance for a very specific area of bio-design automation that desperately needs more contributions (and visible publications)
We appreciate the reviewer's enthusiasm for the need for more visible and accessible bio-automation methods and strongly agree!
 - The experimental results are provided for experimentalists (clearly). That being said, they are largely irrelevant to a programming language. A programming language will enable a set of operations in HW. What someone does with those operations is up to them. The programming language needs to be efficient, sufficiently abstract, expressive, and portable. None of that is the focus of the paper. I totally understand the challenge the authors have in this space trying to get this into a venue read by biologists and not computer scientists. It would be nice to see a more programming oriented white paper or something similar made available to the python programming community that complements this work.
We thank the reviewer for recognizing the challenge of presenting the platform in a way that is compelling to intended (biologist) users. We hope to provide additional documentation and possibly a white paper in the future, but to address the reviewers comments, we have included substantially more detailed Pyhamilton standard python documentation as supplemental information to the manuscript, and updated the github accordingly (as stated above).
-

Reviewer #2:

Summary: In this manuscript the authors successfully expanded the experimental capabilities of robotic liquid handlers by developing a new, open-source, flexible Python software platform for Hamilton STAR, STARlet and vintage robots. The package, Pyhamilton, makes programming and running advanced methods on Hamilton robots much more straightforward, and also enables physical robot motions not allowed in the manufacturer software. Such programming enables more complex experiments, which the authors demonstrate with a growth screen of E. coli BL21 under a large number of different carbon, phosphorus, and nitrogen levels.

Although the developed package solves an existing problem with automating experimental assays in Hamilton platforms, a major limitation is that it appears to be limited to only Hamilton products. As such, there does not appear to be any broadly applicable new capabilities which can be ported to other liquid handling platforms to increase the impact of the code base and demonstrated methodologies. In line with this, it brings to question the level of development of Hamilton's software and if there are similar limitations from other lab automation hardware providers (such an evaluation was not provided).

Major comments

1. Given that the overall work scheme is targeting the Hamilton robotic platforms only, it is suggested to change the current paper's title to imply the same. There is no mention of applicability of the work to other liquid handling / automated platforms. For example, line 90 states that three is a major limitation of the existing software which makes it no faster than hand-pipetting. Is this true of similar platforms to Hamiltons products?

In general, today's automated pipetting robotic systems are slower than an experienced experimentalist on a per-liquid-transfer basis, regardless of robotic platform. The advantages of automation lie not in the speed of execution, but in the repeatability of protocols, especially when they require so many repetitions as to fatigue a technician, and in the reduction in labor from performing two or more operations at once. For example, no ordinary laboratory procedure for a human experimentalist would ask for different volumes for each of the 96 wells within 10 96-well plates (as we detail in the Population Dynamics section), because this cannot be accomplished with a multi-channel pipette, the cognitive load on the experimentalist would be huge for completing the task with a single pipette, and error rate would be high. A robotic platform with 8 independently controlled channels can finish this task more efficiently, even though it takes longer than a multi-channel to transfer each column. When the task is simpler, and it could be done with a multi-channel pipette, the robot can only compete with a fast scientist by addressing multiple plates simultaneously, or incorporating peripheral software (such as with the Turbidostat method). There are signs that this may be changing, like the near-human speed of the small, cheap OpenTron OT2, but to the best of our knowledge, the larger long-lived robots from market leaders such as Hamilton and Tecan are quite a bit slower than a human, but substantially more capable of advanced tasks.

With respect to other platforms, since Pyhamilton uses a platform-independent, web-based protocol (HTTP) and common readable data format (JSON) to bridge Python and HSL, the package can be ported to other platforms by writing an interpreter for the new platform that can send and receive network GET and POST requests, a ubiquitous feature set. We have added a supplemental figure (Supp. Figure 1: Layers of Abstraction in Pyhamilton) and the following sentence to the discussion: *"Presently, Pyhamilton is only extensible to Hamilton robots. However, since it uses a platform-independent, web-based protocol (HTTP) and common readable data format (JSON) to bridge Python and the Hamilton Scripting Language (HSL) (Supp. Figure 1), Pyhamilton could be ported to other biological automation systems that provide an API, such as Tecan or alternative platforms."*

2. The authors demonstrated that they were able pipetting liquids over two 24-well plates simultaneously and thus enabling the robot's full mechanical capability. Can the same procedure be applied to 96-well plates as well and other standard plate formats? Please address the scope of applications that this new capability addresses. Along these lines, can the authors comment on if it is possible to program in any motion that is physically possible on the Hamilton robot with the code base? Providing more examples or a list of expanded functionalities would be ideal, if possible. Line 339 hints at an expansive list.

Yes, The same procedures can be applied to 96-well plates, 6-well plates etc, in any new method, so long as the experimenter defines the plate format in the Hamilton Layout file. One additional advantage of Pyhamilton is that since it interfaces seamlessly with the Hamilton Run control software, any hardware item can be integrated so long as the aspirate, dispense, plate transfer, etc, steps are tested in a new method. For example, we are able to easily add new types of plates (round bottom/flat bottom/deep well) etc, since their dimensions are standardized and easily integratable into layout files. We have added links to each layout file in Supplemental Table EV2 along with additional video recordings and example Hamilton Simulations to further clarify these possibilities. Though we would like to be able to ultimately provide a white paper on Pyhamilton (as later suggested by Reviewer #1), we have included an updated README file as well as a supplemental document in standard Python Documentation

format which thoroughly details the class categorizations used in each method with substantially more detail than we could provide in the manuscript alone. This documentation is also available at: <https://dgregton.github.io/pyhamilton-docs/>. We thank the reviewer for requesting this documentation as we agree that it is critical for proper sharing and dissemination--- our primary motivation behind the development of Pyhamilton in the first place.

3. Line 180. There is a general lack of details for the described turbidostat functionality and how pyHamilton makes this functionality uniquely possible. How were the bacterial turbidostats "maintained"? Specifically, it seems warranted to describe the cultivation conditions in more detail in the main text such as mixing and aeration control, volumes, the range of interventions (adding media, sampling from a given well) possible with the number of replicates, etc. Denoting the specific actions that are impossible with the available manufacturers code vs. pyhamilton seems to be an opportunity to showcase the utility of the code base. Again, is this functionality usable with other liquid handling platforms?

In principle, one could use Hamilton Scripting Language (HSL) to enable very basic turbidostat functionality, but doing so would likely be beyond the capabilities of even most biologists who work with robots, and it would not permit the maintenance of so many turbidostats at once. That is, the asynchronous method execution enabled by Pyhamilton is required to support a large number of turbidostats that correctly respond to absorbance and fluorescence results from the plate reader. We would prefer to keep the details of cultivation conditions that we detailed in the Methods section (specifically in the "Feedback controller algorithm" section) and as described in Figure 2a, as they don't seem directly relevant to understanding the functionality afforded by Pyhamilton. Additionally, we have added clarification to the "High-throughput turbidostat cultures" methods section. We have addressed the question of other platforms above. We have also added Supp. Figure 5, detailing the average number of volumes/hour and growth rates for various types of media (M9, 2XYT, LB), and simulations of the maximum limitations of the method (# turbidostats v. max growth rate, Supp. Fig 6) so that biologists can adapt the number of plates necessary for their respective applications. The reviewer is correct in that the mixing/media demands will vary according to each respective experiment, which must be validated experimentally. We hope that the information we have provided on timelines/volume usage sufficiently addresses the reviewer's comment.

4. It was mentioned several times that Pyhamilton makes programming more complex robotic methods much easier than in the manufacturer's software, and enables the programming of methods that are impossible in the manufacturer's software (such as the feedback loop used to control the OD in the turbidostat method). It would benefit readers who are not familiar with Hamilton products (which is probably the vast majority of the readership) to include a brief description of why this is so. This description could be made stronger by including an example script of the Hamilton control software and what the limitations are for the closest best option without pyhamilton (if this is allowable by their terms of service).

We thank the reviewer for the excellent suggestion. We have added additional documentation to clarify the difference, most notably Supplementary Video 1, in which we program a simple task (moving tips from full to empty tip boxes in a staggered pattern) using manufacturer software and Pyhamilton side-by-side.

5. At the Github pyHamilton's repository authors mentioned "... on Windows XP and Windows 7". What is required to run the software implementation on the most recent Windows and their dependencies, as these versions of Windows are obsolete and unusable on most institutional networks?

As of trials between our last submission and now, Pyhamilton has been tested in simulation on Windows 10. Since this test interfaced with official Hamilton executables to run the simulation, we are fairly certain that it would work with a physical robot attached as well, though this is not possible to test for us at the moment without an

additional robot or reconfiguration. To our knowledge, no research groups are using Hamilton robots on a Windows operating system more recent than Windows 7 at this time, but a collaboration with a group that does would be a great help in ensuring the forward compatibility of Pyhamilton.

6. It is unclear what extent of human intervention is necessary when running the robots using pyHamilton for the presented test cases? These should be included in each example to provide context. Initial set up and interventions during run time.

We thank the reviewer for the excellent suggestion. We have added a new Supplemental Table EV2 which contains information on personnel time, set-up time, iteration time, the relevant consumables, robotic equipment, suggested manufacturer and part number, estimated cost per item/experiment, and githubs links to each example script, robotic layout file, and a video of the respective method and simulation run.

7. For designing/testing and running experiments, is it possible to simulate a given design/experiment before each run? I.e., is there a simulation mode or if not, why no?

We have further clarified that all Hamilton methods can be easily run in simulation mode with Hamilton Run control software (using a clarifier "HamiltonInterface(simulate=True)"). This is and has been critical in the development of all methods detailed in the manuscript and each new method we use in the laboratory. We apologise for not highlighting this more clearly in the text and have adjusted the manuscript accordingly. We have also added screen recordings of simulation mode runs for each of the methods documented in the manuscript, which can be found in Supp. Videos 2-4.

8. Section (High-throughput perturbation analysis of metabolites), it can be assumed that the authors used strains with more or less similar growth rates. Changing media components to optimise expression and growth is not challenging using the system presented, however screening strains that differ in their growth rates would be challenging if such screens were run at the same time. Please address the applicability of the current code base to such a setup of using strains with different growth rates.

To address this question, we performed additional experiments in which we grew the same strain in four different media formulations which yield varying growth rates (Supp. Figure 5). So long as the robot could dilute the fastest-growing culture quickly enough, handling additional cultures that grow more slowly proved a minimal additional burden. However, the reviewer is absolutely correct that due to the increased medium requirement of faster growing strains, there is a trade off between hands-on-time, growth-rate, and the number of cultures that can be maintained at any given time. Thus, we have added an additional supplemental figure simulating the maximum limitations of the turbidostats (growth rates v. max # turbidostats, Supp. Fig 6), and respective media volume requirements. Though we do not observe that the method itself is hindered substantially by varied growth rates (as this was a design goal implemented in the section involving 25 different growth media conditions), the hands-on-time does vary from experiment-to-experiment and when the bacteria are growing too quickly for an experimenter to keep up with refreshing media (minimum user intervention time is once per 10 hours at max growth rates). Thankfully, with Pyhamilton, error handling, notification, and user communication can be readily implemented into any method and easily interfaced with peripheral data analysis (such as R) in order to notify the user remotely that their ongoing experiment requires intervention. We do expect that biologists implementing this protocol will have to optimize the method for their individual needs, and have documented that in the text.

Minor comments

1. The cross-contamination tests were run for 24 hours. This seems like a rather limited time duration and should be put in that context.

We have emphasized that the finding occurred over this duration, and to further satisfy the reviewer, we have added Supplemental Figure 6 as described above which includes a 96-well plate run in which turbidostats were maintained over 48 hours and the total amount of volume used over that time was quantified. We additionally quantified back-contamination into the media, which was found to be negligible. We also hope that by easily sharing these protocols, that new users will be able to easily implement additional peripherals that suit their needs and we look forward to the many possible methods that other diverse research groups may implement in the future (e.g. we have been able to implement additional tip-reuse washing steps for protocols that require more bleaching time by disposing of bacteria in washers controlled by off-deck pumps. This methodology is detailed in Python Documentation, but is beyond the scope of the manuscript).

2. A lot of interesting methods were developed for the specified robotic platform. It would be of great help in order to replicate the work for other researchers to include small video clips to each developed method, if possible.

We thank the reviewer for the suggestion and have added video clips for each of the methods.

3. Line 545: It isn't entirely clear from this description how Pyhamilton interacts with the Hamilton liquid handling robots. Is the manufacturer's software running at the same time, or does this software interact directly with the robot?

The user's and programmer's experience with Phyhamilton is that the python script runs, and the robot moves, with no other visible software launched on screen. Two exceptions are when the user specifies that the Hamilton interface should run in simulation mode, which launches the manufacturer's Hamilton Run Control (using a clarifier "*HamiltonInterface(simulate=True)*"), or when the robot requires using plate reader software, which runs in the background. "Behind the scenes," both the regular and simulated execution modes launch the same Hamilton-developed HSL executable interpreter via different means. We have added a schematic of the pyamilton architecture, Supp. Fig 1. to further clarify this.

4. What was the trace elements recipe, line 665?

We have clarified the details in the methods section and apologize for the oversight.

5. Fig. 1A/C/G/J: Is this font required? If not, it would be beneficial to change it to something easier to read. We have modified the font for clarity.

6. Fig. 1A: The references to Fig 1C, 1F, 1H should probably read 1C, 1G, 1J

We have clarified these references.

7. Missing axis label for figure 2d, please add.

We apologize, and have corrected this error.

8. Short description of the robotic method, corning (line 673) Item#3631, is required.

We have clarified this item.

9. Line 255: Should "All cultures grew within +/- 20% of M9 media growth rate" read "20% of the standard M9 media growth rate"?

We have clarified this text.

10. Fig 3C: You should have the phosphorus concentration levels labelled as in 3B, since the line implies a continuous gradient of phosphorus and gives less information on the concentration differences between the plots. We have updated this figure.

11. Can you please provide a reference for 2XYT medium used, Line 558?

We have provided a reference for 2XYT media.

Here is a list of grammatically incorrect and misspelling statements that needs a careful review beside a thorough edit for more clarity and grammar check:

We thank the reviewer for their incredibly thorough review, have clarified or corrected all the following errors, and apologise for the oversights.

1. E. coli should be italic or underlined throughout the text.
2. Typo in media medium for singular, line 674.
3. Line 88: Comma needed after simultaneously
4. Line 135: shouldn't read "the python"
5. Fig. 1B caption: allow for improve throughput of laboratory across 24-well plates should be rephrased
6. Line 241: The paragraph break at the end of this sentence should be reconsidered, as the description of the experimental setup is split between the paragraphs.
7. Line 563: Bacterial should read bacteria
8. Text under figure 1, the following sentence needs rephrasing? "(C) ... bacteriophage plaque assay show".
9. Figure 2, missing 1a?
10. 10. For cultures of S2060 (Line 558), Add the Addgene reference number. Is this (Bacterial strain #105064) correct?

Reviewer #3:

The authors develop an open-source Python platform with the goal of improving the flexibility of Hamilton liquid handling systems and enabling high-throughput experiments of greater complexity. Their system allows for pipetting steps to be coupled in real time to read outs from a standard microplate reader, enabling the setup of controllers that dynamically respond to the current state of the experiment. The authors apply their platform to a number of different test cases centered around cell growth. Their main examples focus on functionally converting multiple well-plates into turbidostats with cultures that are maintained at a fixed OD. They further utilize this capability to study the impact of carbon, nitrogen, and phosphorus sources on both cell growth and protein production in defined media. Additionally, they develop and implement a cleaning process that allows for the re-use of pipette tips.

This work represents a significant improvement to the capabilities of standard tip-based liquid handling systems. The implementation of controller modules that can continuously modify reaction conditions in a well-plate is particularly exciting, and the example of converting standard plates into turbidostats is both convincing and impactful. Their systematic interrogation of the impacts of limiting nutrients on growth rate and protein expression yields highlights the flexibility of the system and provides insight into the metabolic processes of bacteria under resource limited growth conditions. The manuscript is well written, and the examples are presented in a clear and concise manner, with one exception noted below.

Major comments

- The purpose and meaning the population dynamics simulation in figure 1 d-g is not clear. Additionally, the treatment of the topic in the text is cursory. It seems that the purpose is to show the ability to pipette different quantities of liquid in complex and programmable patterns. This is obscured by the example of population dynamics given the instrument is pipetting dye. A more detailed discussion of how this relates to experimentally testing population dynamics is warranted.

Our goal was to explain the capabilities afforded by complex pipetting to an audience of experimental biologists in a compelling manner. While we agree with the reviewer that transferring whole organisms rather than dye would be more compelling, these hypothetical experiments involve studying mating type patterns (such as conjugative *E. coli*, mating type yeast, or dioecious nematodes) and are beyond the scope of the manuscript. We found that this example was especially compelling to the many biologists we consulted, and that using dye most clearly demonstrated a type of previously inaccessible experiment that could be performed readily with Pyhamilton. We believe that the discussion of possible applications is sufficiently described by the section titled: "Enabling liquid transfers requiring complex calculations", within the text: "*Succinct code (Fig. 1G) can generate both symmetric and asymmetric diffusion patterns, which could be combined with a phenotypic reporter to experimentally simulate arbitrarily directionally bounded or unbounded migration (Fig. 1D) with many model organisms such as E. coli, yeast, or even nematodes.*"

- The data associated with the cleaning protocol in figure 2 c only show that no cross contamination between wells occurred, not that the media source wells were not contaminated. This should also be proven.

We have performed additional experiments to examine this question and observed no contamination of the media source wells, which is detailed in Supplemental Figure 5. We also maintained cultures over 48 hours with similar results (Supp Fig 6). We hope this satisfies the reviewer's concern.

Minor comments

- The limitations of the system are not discussed in detail.

We have modified the discussion section to include the limitations of the present system's applicability to only Hamilton Robots, and proposed how one might extend the methodology to other systems as follows:

"Presently, Pyhamilton is only extensible to Hamilton robots. However, since it uses a platform-independent, web-based protocol (HTTP) and common readable data format (JSON) to bridge Python and the Hamilton Scripting Language (HSL) (Supp. Figure 1), Pyhamilton could be ported to other biological automation systems that provide an API, such as Tecan or alternative platforms."

- In the case of the plate-based turbidostats what are the limitations on the growth rate of the organism that can be maintained, both within a single experiment and alone? What are the maximum number of

cultures that could be maintained in a given experiment? Is this a limitation of the footprint available for the pipetting system, or a limitation of the transfer speed?

As also described in our response to reviewer 2 above, we performed additional experiments in which we grew the same strain in four different media formulations which yield varying growth rates, which we have included in Supp. Figure 5. So long as the robot can dilute the fastest-growing culture quickly enough, handling additional cultures that grow more swiftly required minimal additional operation time. However, the reviewer is correct that the increased medium requirement of faster growing strains creates a trade off between hands-on-time, growth-rate, and the number of cultures that can be maintained at any given time. The method itself is not at all hindered by varying growth rates (as this was a design goal implemented in the section involving 25 different growth media conditions), but the hands-on-time will vary from experiment-to-experiment and must be determined empirically for each new run. Thankfully, with Pyhamilton, error handling, notification, and user communication can be readily implemented into any method in order to notify the user remotely that their ongoing experiment requires intervention. We do expect that biologists implementing this protocol will have to optimize the method for their individual needs, and have documented that in the text.

- One of the major limitations of tip-based robotic systems is the long setup times for complex transfer protocols. What is the timescale for performing several hundred transfers, as was done in the population dynamics example, and is this significant on the timescale of the experiment.

The reviewer is correct that liquid transfers are a rate-limiting step in all tip-based robotics systems, including Hamiltons. As we describe, using Pyhamilton to implement asynchronous operations helps with this, as the robot can simultaneously accomplish other tasks such as using the plate reader while pipetting. We have added time-stamped recordings of each of the methods in hope that the added videos and simulations for each of the methods clarifies the timescale requirements for the end users, as they can vary from very short (~15 min for the plaque assay, 20 min/plate for the turbidostats) to very long (2 hours for hundreds of transfers during the population dynamics experiment). We have also documented these details more thoroughly in Supplemental Table EV2.

Thank you for sending us your revised manuscript. We have now heard back from the two reviewers who were asked to evaluate the revised study. As you will see below, both reviewers are satisfied with the modifications made and are supportive of publication. Reviewer #2 raises a remaining minor concern, which can be addressed by text edits and we would ask you to fix it in a minor revision.

Before we can formally accept the study for publication, we would also ask you to address some pending editorial issues listed below.

REFEREE REPORTS

Reviewer #1:

The authors have sufficiently addressed my comments from the previous review. I think the paper is now suitable for publication. This has been done primarily via supplemental text and videos.

However, key sentences have been added to the main text that I think add the additional emphasis and information needed.

Reviewer #2:

- One thing that probably should be clarified is the actual numbers of experiments which can or have been simultaneously ran. Perhaps there is rounding, but these should be consistent and the non-rounded values are likely best. For example, is it 500 or 480 (both numbers given)? Similarly, is it 96 or 100?

- The videos were a nice addition.

- We feel like the concerns from the original critique have been satisfied

Reviewer #3:

The authors have addressed my concerns.

The authors have made all requested editorial changes.

Accepted

5th Feb 2021

Thank you again for sending us your revised manuscript. We are now satisfied with the modifications made and I am pleased to inform you that your paper has been accepted for publication.

Corresponding Author Name: Kevin Esvelt

Manuscript Number: MSB-20-9942